# Crosstalk between Methylation and ncRNAs in Breast Cancer: Therapeutic and Diagnostic Implications

**DOI:** 10.3390/ijms232415759

**Published:** 2022-12-12

**Authors:** Yitong Liu, Ping Leng, Yan Liu, Jinlin Guo, Hao Zhou

**Affiliations:** College of Medical Technology, Chengdu University of Traditional Chinese Medicine, Chengdu 610000, China

**Keywords:** breast cancer, ncRNAs, epigenetic, therapeutic, diagnostic strategy, DNA methylation, N6-methyladenosine, histone methylation

## Abstract

Breast cancer, as a highly heterogeneous malignant tumor, is one of the primary causes of death among females worldwide. The etiology of breast cancer involves aberrant epigenetic mechanisms and abnormal expression of certain non-coding RNA (ncRNAs). DNA methylation, N6-methyladenosine(m6A), and histone methylation are widely explored epigenetic regulation types in breast cancer. ncRNAs are a group of unique RNA transcripts, mainly including microRNA (miRNAs), long non-coding RNA (lncRNAs), circular RNA (circRNAs), small interfering RNA (siRNAs), piwi-interacting RNA (piRNAs), etc. Different types of methylation and ncRNAs mutually regulate and interact to form intricate networks to mediate precisely breast cancer genesis. In this review, we elaborate on the crosstalk between major methylation modifications and ncRNAs and discuss the role of their interaction in promoting breast cancer oncogenesis. This review can provide novel insights into establishing a new diagnostic marker system on methylation patterns of ncRNAs and therapeutic perspectives of combining ncRNA oligonucleotides and phytochemical drugs for breast cancer therapy.

## 1. Introduction

The latest global statistics in 2020 indicate that breast cancer (BC) surpassed lung cancer to become the most commonly diagnosed cancer globally [1]. The pathogenesis and progression of BC include both genetic and epigenetic factors. Epigenetic instability plays an essential role in BC occurrence and development [2]. In epigenetics, DNA sequences do not change but gene expression undergoes heritable alternations, and epigenetic regulation mainly includes DNA methylation modification, histone covalent modification, genomic imprinting, and chromatin remodeling [3]. Such subtle epigenetic modulation is very important for genome stability while dysregulated epigenetic mechanisms might lead to breast tumor initiation [4,5].

### 1.1. DNA Methylation

DNA methylation refers to the process where a methyl group from S-adenosyl methionine (SAM) is covalently and reversibly added to the fifth carbon position of the cytosine ring of the genomic CpG dinucleotide under the catalytic action of DNA methylation transferase (DNMT) [6] (Figure 1). DNA methylation controls gene expression and genome stability [7]. Generally, DNA hypermethylation inhibits gene expression while DNA hypomethylation activates gene expression. Aberrant DNA methylation directly or indirectly alters the transcription of tumor suppressors and carcinogenesis factors, thereby accelerating the development of BC [8,9].

### 1.2. N6-Methyladenosine (m6A)

N6-methyladenosine (m6A), methylation at the N6 position of adenosine, is the most frequent modification of mRNA, and non-coding RNAs are also highly modified with m6A [10,11] (Figure 1). This dynamic and reversible post-transcriptional RNA modification is mediated by methyltransferases (commonly called writers), demethylases (commonly called erasers), and recognition proteins (commonly called readers) [12]. The alterations of m6A levels may affect RNA processing, degradation, and translation, disrupt gene expression and lead to tumorigenesis [10,13]. Niu et al. demonstrated that demethylase fat mass and obesity-associated (FTO) mediates m6A demethylation in the 3′ UTR of tumor suppressor BCL2 Interacting Protein 3 (BNIP3) mRNA and silences BNIP3 to promote BC progression [14].

### 1.3. Histone Methylation

Histones are the core components of the nucleosome subunits in which histones H3, H4, H2A, and H2B form an octamer surrounded by DNA fragments [15] (Figure 1). Histone methylation is a chemical modification that covalently occurs on the arginine and lysine residues of H3 and H4, which may be monomethylation, dimethylation, or trimethylation [16]. Histone H3 at lysine 4 (H3K4), histone H3 at lysine 36 (H3K36), and histone H3 at lysine 79 (H3K79) methylation modifications are all involved in transcriptional activation, whereas histone H3 at lysine 9 (H3K9), histone H3 at lysine 27 (H3K27), and histone H4 at lysine 20 (H4K20) methylation have been shown to be linked to transcriptional silencing [17]. Histone methylation disorders promote BC tumorigenesis [18]. Jeong et al. revealed that the catalytic activity of histone methyltransferase nuclear SET domain-containing protein 3 (NSD3) toward H3K36 methylation induces dimethylated or trimethylated histone H3 at lysine 36 (H3K36me2/3) dependent transactivation of genes connected with NOTCH signaling, driving breast tumor initiation and metastasis by the NSD3/NOTCH signaling regulatory axis [5].

### 1.4. Non-Coding RNAs (ncRNAs)

Non-coding RNAs (ncRNAs) are generally comprised of microRNA (miRNAs), long non-coding RNA (lncRNAs), circular RNA (circRNAs), small interfering RNA (siRNAs), piwi-interacting RNA (piRNAs), etc. [19]. MiRNAs are small non-coding RNAs ordinarily with a length of about 20 nucleotides, serving as paramount regulators of gene expression [20]. MiRNAs and Argonaute protein can form a transcriptional silencing complex in which seven or eight nucleotides in the seed region at the 5′ end of miRNAs are complementarily paired with the 3′ untranslated region (3′ UTR) of the target mRNA [21]. MiRNAs often act as post-transcriptional regulators to inhibit or degrade the target mRNA, resulting in gene silencing [22]. LncRNAs are long non-translated RNAs with more than 200 nucleotides in length that are transcribed by RNA polymerase II and distributed in the cytoplasm and nucleus, thus performing diverse biological functions [22]. CircRNAs are new closed-loop untranslated RNAs that lack the 5′ cap and 3′ terminal poly-A tail, forming a closed loop structure through back-splicing and head-tail bonding, acting as miRNA sponges, and having multiple miRNA-specific binding sites [23,24]. Endo-siRNAs are 20–24 nt endogenous small interfering RNAs that can inhibit abnormal perturbations of retrotransposable elements in the genome through RNA interference and negatively modulate gene expression at the post-transcriptional level [25]. PiRNAs are small non-coding RNAs different from miRNAs and siRNAs, approximately 26–30 nt in length, which participate in the silencing of transposable elements [26]. Dysregulation of ncRNAs presumably promotes cancer initiation and progression [27]. For example, Gao et al. found that lncRNA PTENP1 acts as an miR-20a endogenous sponge to positively regulate the gene of phosphate and tension homology deleted on chromosome ten (PTEN) expression, inhibiting BC progression by the PTENP1/miR-20a/PTEN axis; the deregulation of lncRNA PTENP1 can promote BC progression [28].

The ncRNA expression and function can be modified and controlled by methylation modification alteration, while the methylation may also be efficiently regulated by ncRNAs. In this review, we minutely describe the intricate interaction between methylation modifications and ncRNAs in BC. Furthermore, we summarize the diagnostic strategies for detecting the methylation status of ncRNAs in peripheral blood (PB) and propose new therapeutic ideas based on methylations and ncRNAs.

## 2. Crosstalk between Methylation and ncRNAs in BC

### 2.1. MiRNA

DNA methylation, m6A, and histone methylation modulate miRNA expression and stability in BC. Moreover, miRNAs may influence the methylation-related enzyme components at the post-transcriptional level to regulate DNA methylation, m6A, and histone methylation as coordinated regulations. The destruction of interactions between methylation modifications and miRNAs may lead to BC progression. This section mainly introduces the crosstalk between different methylation types and miRNAs in BC. Key findings in relevant studies are condensed in Table 1.

#### 2.1.1. Crosstalk between DNA Methylation and miRNA

miRNA regulation by DNA methylation

DNA methylation modulates miRNA expression where high methylation of CpG islands in the promoter inhibits miRNA transcription while low methylation of CpG islands activates miRNA expression [68] (Figure 2A). The miRNA expression dysregulation caused by aberrant DNA methylation patterns may disrupt diverse physiological or pathological processes such as proliferation, apoptosis, metastasis, stemness, drug resistance, and tumor microenvironment, leading to the malignant progression of BC (Table 1). Kang et al. demonstrated that miR-362-3p and miR-329 are downregulated by promoter DNA hypermethylation and inhibit cell proliferation, migration, and invasion by negatively regulating p130Cas in BC [29]. Notably, Poli et al. revealed that the methylation level of basal BC cells is higher than luminal subtype cells, and the miR-29c expression is negatively relevant to the methylation status, which indicates that the differential methylation pattern drives the subtype-specific expression of miRNA [48]. Interestingly, Gong et al. evaluated the DNA methylation patterns of BC subtype-related miRNAs among American women of African ancestry and European ancestry and found differential methylation sites for lineages, suggesting that the DNA methylation patterns of miRNAs are not only subtype-specific but also lineage-specific [69]. In addition, circadian rhythm disturbances and xenoestrogen exposure may transform the DNA methylation pattern of miRNA and lead to epigenetic genetic alterations in BC [70,71,72]. The detailed diagram of miRNA regulated by DNA methylation is shown in Figure 2A.

2.Regulation of DNA methylation by miRNA

DNA methylation is maintained and established by members of the DNMT family, DNMT1, DNMT3A, and DNMT3B [73]. DNMT acts on the CpG island of the miRNA host gene promoter to regulate methylation status, thereby affecting miRNA expression. In turn, miRNAs can target binding to DNMT mRNA, silencing DNMTs mRNA at the post-transcriptional level and subsequent DNMT protein synthesis reduction [55] (Figure 2B). Obviously, the methylation state of the miRNA host gene promoter is regulated by DNMT, and miRNAs can target and negatively regulate the expression of DNMT, which can form a miRNA/DNMT regulatory circuit involved in the initiation and development of BC (Figure 3). Pang et al. demonstrated that hypermethylation of the promoter CpG island mediated by DNMT3A represses miR-200b, and miR-200b can negatively regulate DNMT3A expression, thus forming a feedback loop in triple-negative BC [57]. It is significant to note that miRNAs targeting DNMT is performed in the cytoplasm while the DNA methylation modification of miRNA host genes by DNMT is performed in the nucleus (Figure 3).

#### 2.1.2. Crosstalk between m6A and miRNA

M6A modification is involved in miRNA biogenesis, and abnormal m6A modification on miRNAs can worsen tumor progression [74] (Figure 2C). Pan et al. revealed that methyltransferase-like 3 (METTL3) accelerates miR-221-3p maturation and enhances miR-221-3p expression by increasing m6A modification of pri-miR-221-3p and promotes the drug resistance of BC cells to Adriamycin via the METTL3/miR-221-3p/HIPK2/Che-1 axis [61]. METTL3 regulates miRNA processing and maturation through m6A modification. Therefore, dysregulation of this process may promote BC progression and chemoresistance. The paradigm of m6A modification on miRNA is shown in Figure 2C.

#### 2.1.3. Crosstalk between Histone Methylation and miRNA

miRNA regulation by histone methylation

Histone methylation markers (H3K9me2, H3K9me3, and H3K4me3) regulate miRNA expression via modulating chromatin accessibility to transcription (Figure 2D). Simonini et al. demonstrated that the high expression of miR-375 in ER-positive BC cells is partly attributed to the loss of the epigenetic mark H3K9me2 [62]. Damiano et al. found that Zinc-Finger E-Box Binding Homeobox transcription factor ZEB1 regulates H3K9me3 and participates in the epigenetic silencing of miR-200c, which is connected to the aggressiveness of BC [63]. Similarly, Mitra et al. uncovered that histone demethylase Jumonji/ARID1B (JARID1B) binds to the let-7e promoter region, removing the chromatin marker H3K4me3 that is associated with transcriptional activation and promotes BC cell proliferation by epigenetically inhibiting let-7e expression [64]. These studies have brought important insights into the complicated interrelationships between histone methylation modifications and miRNAs in BC progression. A detailed description of miRNA regulated by histone methylation is summarized in Figure 2D.

2.Regulation of histone methylation by miRNA

Regulation of histone methylation by MiRNAs is a crucial mechanism in BC (Figure 2E). For example, Yu et al. proved that miR-7 targets H4K20 specific monomethyltransferase SET domain-containing protein 8 (SET8) and inhibits H4K20 monomethylation by down-regulating SET8 to suppress BC cell invasion [65]. Hui et al. reported that miR-491-5p inhibits tumorigenesis and progression in ER-positive BC by targeting histone demethylase JMJD2B [66]. Similarly, Wu et al. found that miR-29a targets H4K20-specific trimethylaterase SUV420H2 and promotes EMT, migration, and invasion in BC cells [67]. These studies highlight the opinion that miRNAs are implicated in regulating histone methylation by targeting methyltransferase or demethylase to accelerate BC progression. The precise mechanism is illustrated in Figure 2E.

### 2.2. LncRNA

The expression of lncRNAs can be adjusted by epigenetic mechanisms covering DNA methylation, m6A, and histone methylation modification. Conversely, some lncRNAs can modulate DNA methylation, m6A, and histone methylation modification patterns by recruiting chromatin modifiers to form multifunctional gene regulatory structures. The dysregulation between different methylation modification types and lncRNAs can lead to BC, and we mainly discuss the interaction between methylation modifications and lncRNAs in BC. Related findings are summarized in Table 2.

#### 2.2.1. DNA Methylation and lncRNA

lncRNA regulation by DNA methylation

Similar to the regulation of miRNA by DNA methylation, the high methylation level of the lncRNA host gene promoter CpG island suppresses lncRNA expression and the low methylation level promotes expression (Figure 2F), which suggests that DNA methylation is negatively correlated with lncRNA expression. Decreased DNA methylation levels leading to oncogenic lncRNA expression upregulation, and increased DNA methylation levels leading to antioncogenic lncRNA downregulation can both promote the malignant behavior of BC. Wang et al. reported that lncRNA EPIC1 is a carcinogenic nuclear lncRNA that is upregulated due to promoter CpG island hypomethylation and promotes cell cycle progression in BC [76]. Similarly, Zheng et al. elaborated that lncRNA HUMT is upregulated by promoter hypomethylation, promoting lymphangiogenesis and metastasis by activating Forkhead box k1 (FOXK1) in triple-negative BC [77]. In addition to promoter CpG island methylation modification, intergenic CpG island methylation modification also plays a significant biological role in regulating lncRNA expression. For example, Lu et al. found that lncRNA HOTAIR shows a positive correlation with downstream intergenic CpG island methylation, which is different from the negative effect of promoter DNA methylation [103]. The detailed diagram of lncRNA regulated by DNA methylation is shown in Figure 2F.

2.Regulation of DNA methylation by lncRNA

LncRNA can mediate DNA methylation alterations during the development of BC. First, lncRNAs can direct DNA methylation by recruiting DNMTs to specific sites [104] (Figure 2G). Wang et al. demonstrated that LINC00518 recruits DNA methyltransferases to promote the methylation of CDX2 and activate the Wnt signaling pathway to regulate BC epithelial cell growth and metastasis [83]. Moreover, lncRNA can adjust the binding status of DNA methyltransferase and the target gene promoter region by controlling the SAH accumulation to influence the DNA methylation state (Figure 2H). Wang et al. uncovered that lncRNA H19 affects the binding of DNMT3B and the Beclin1 promoter region by altering the SAH accumulation subsequently downregulating the methylation of the Beclin1 promoter and promoting tamoxifen resistance and autophagy via the H19/SAHH/DNMT3B axis in BC cells [84]. Notably, lncRNA can form RNA-protein modification complexes with methyltransferases to modulate DNA methylation. Li et al. found that adaptor protein LLGL2, lncRNA MAYA, and methyltransferase NSUN6 form an RNA-protein complex that methylates MST1 at Lys59, resulting in MST1 kinase inactivation and YAP target gene activation, thereby triggering BC osteoclast differentiation and bone metastasis [80]. Besides, lncRNAs can function as cis-acting elements to modulate DNA methylation. Xu et al. revealed that lncRNA MAGI2-AS3 acts as a cis-regulatory element to downregulate DNA methylation in the promoter region of MAGI2 to inhibit BC progression [90]. In addition, the H19/IGF2 locus is mainly modulated by the methylation status of the imprinting control center [81]. Vennin et al. found that lncRNA 91H regulates the expression of the H19/IGF2 imprinting locus by masking the methylation site on the imprinting control center and the H19 promoter, which plays a crucial role in promoting the aggressive phenotype of BC [81].

#### 2.2.2. m6A and lncRNA

The m6A regulatory protein METTL3 is involved in the lncRNA m6A modification, and lncRNA as the master regulator can also modulate m6A RNA methylation of the downstream target gene. Specifically, the m6A modification on lncRNA mediated by METTL3 can regulate the lncRNA sponge’s competitive endogenous RNA (ceRNA) network activity (Figure 2I). Fan et al. reported that METTL3 increased the m6A methylation modification of LINC00675 that acts as a ceRNA for miR-513b-5p and could repress miR-513b-5p expression, this m6A modification did not affect the LINC00675 expression but enhanced the interaction between LINC00675 and miR-513b-5p [93]. In addition, lncRNA could recruit m6A methyltransferase and participate in the m6A modification of the downstream target gene that regulates BC progression (Figure 2J). Sun et al. [92] uncovered that LNC942 recruits methyltransferase-like 14 (METTL14) as an oncogene and increases the downstream target genes CXCR4 and CYP1B1 expression and stability by promoting post-transcriptional m6A methylation, thereby promoting BC cell proliferation and malignant progression.

#### 2.2.3. Histone Methylation and lncRNA

Regulation of lncRNA by histone methylation

Histone methylation marks on lncRNA are associated with transcriptional activation or repression of lncRNA. For instance, Pang et al. demonstrated that ZNF217 recruits EZH2 to the lncRNA EPB41L4A-AS2 locus and suppresses lncRNA EPB41L4A-AS2 expression by epigenetically increasing trimethylated histone H3 at lysine 27 (H3K27me3) enrichment, thus promoting BC cell proliferation, migration, and invasion [95]. Similarly, epigenetic modifications lead to the upregulation of oncogenic lncRNA DLEU1 partly through increasing histone modifications of H3K4me3 to promote carcinogenesis [94]. These studies indicate that histone methylation mark patterns on lncRNA are profoundly altered in BC. The relevant mechanism is described in Figure 2K.

2.Regulation of histone methylation by lncRNA

LncRNAs regulate histone methylation primarily by recruiting methylation-related modifiers (Figure 2L). Notably, lncRNAs are involved in the recruitment of polycomb proteins that are associated with histone methylation. Zhang et al. reported that LINC00511 interacts with EZH2 and recruits PRC2 to mediate H3K27me3 modification in the promoter region of CDKN1B, contributing to the suppression of CDKN1B expression through an epigenetic mechanism [100]. In addition, lncRNAs recruit methyltransferases to control histone methylation markers of specific targets. The methyltransferase MLL1 is recruited by lncRNA ROR to promote H3K4 trimethylation of TIMP3, thereby enhancing TIMP3 transcription and promoting BC progression [101]. In summary, these findings suggest that lncRNAs exhibit a non-negligible role in regulating the histone methylation marks in BC. The paradigm of lncRNA regulating histone methylation is illustrated in Figure 2L.

### 2.3. CircRNA

CircRNAs also play essential roles in DNA methylation and m6A modification regulation. Gu et al. reported that abnormal DNA methylation of circRNA might lead to the dysfunction of the miRNA sponge, which indirectly affects the expression of miRNA target genes [105]. In addition, emerging studies have demonstrated the role of circRNA serving as the miRNA sponge in modulating m6A RNA methylation. Lv et al. elucidated that circBACH2 serves as the miR-944 sponge, which counteracts the inhibitory effect of miR-944 on the m6A modulator HNRNPC, stimulating the expression of HNRNPC through the MAPK signaling pathway, and promoting BC cell proliferation and progression [106]. Relevant details are summarized in Table 3. The mechanism diagram is depicted in Figure 2M.

### 2.4. Endo-siRNA

Long interspersed nuclear element-1 (LINE-1) is a class of retrotransposable elements [110]. The transcriptional activation of LINE-1 mediated by promoter DNA methylation leads to changes in genomic structure that are strongly associated with cancer progression [111]. Endo-siRNA can modulate the DNA methylation status of the gene promoter region of LINE-1. For example, Chen et al. found a subset of endo-siRNAs is significantly underexpressed in human BC cells, whereas LINE-1 is significantly overexpressed [107]. Mechanistically, siRNA can inhibit LINE-1 expression by inducing hypermethylation of CpG islands in the 5′-UTR promoter region of LINE-1 [107]. This indicates the function of endo-siRNA in maintaining the genome integrity of human cells by inhibiting retrotransposon activity through DNA hypermethylation. Related reports are summarized in Table 3. The detailed mechanism is depicted in Figure 2N.

### 2.5. PiRNA

PiRNA is also involved in the regulation of DNA methylation modification. The aberrant expression of piRNAs may lead to epigenetic dysregulation, contributing to a variety of diseases via altering DNA methylation and chromatin modification [112]. A genome-wide analysis of the roles of piRNAs on gene-specific DNA methylation suggests that single-copy piRNAs may induce DNA methylation alterations at non-transposable element gene loci, partly by directly binding to genomic DNA or nascent mRNA adjacent to target CpG sites [113]. Notably, Fu et al. revealed that piR-021285 is associated with altered 5′-UTR/1 exon methylation of pro-invasive ARHGAP11A, indicating that piRNA participates in gene-specific DNA methylation regulation and affects cell invasiveness in BC [114]. Furthermore, the crosstalk between piRNAs and DNMTs can alter the methylation status of genes. For instance, Liu et al. demonstrated that piRNA-651 and PIWIL2 form a complex to recruit DNMT1 to the PTEN gene CpG islands region, subsequently promoting cell proliferation and migration by downregulating PTEN expression [108]. Similarly, Ding et al. discovered that piR-823 promotes APC gene promoter methylation by increasing the expression of downstream target DNMTs, inducing luminal BC cell proliferation and stemness [109]. Relevant findings are summarized in Table 3. The related regulation mechanism is shown in Figure 2O.

### 2.6. Housekeeping ncRNAs

In addition to regulatory ncRNAs, ncRNAs also include housekeeping ncRNAs such as transfer RNA (tRNAs), ribosomal RNA (rRNAs), small nuclear RNA (snRNAs), and small nucleolar RNA (snoRNAs) [115]. These housekeeping ncRNAs are expressed constitutively and are crucial for maintaining general cell function. tRNAs carry epigenetic modifications which are necessary for tRNA folding, stability, and function [116]. Martinez et al. presented a paradigm that BCDIN3D is a cytoplasmic histidine transfer RNA-specific 5′-methyl phosphate cap synthase which efficiently monomethylates the 5′-monophosphate of cytoplasmic histidine transfer RNA, thus protecting the cytoplasmic histidine transfer RNA transcript from degradation in vitro [117]. In addition, dysregulation of rRNA transcription caused by histone chromatin markers promotes tumor progression. Tanaka et al. demonstrated that glucose starvation induces KDM2A-mediated histone chromatin marker H3K36me2 demethylation at the rRNA gene promoter and decreases rRNA transcription and cell proliferation in BC [118].

## 3. Therapeutic and Diagnostic Implications

### 3.1. Treatment Strategies Based on Methylation of ncRNA

Higher plant constituents show an anticancer effect, and some phytochemicals can exert tumor inhibitory activity through epigenetic regulation [119,120,121,122,123]. Many studies have proved the mechanism of phytochemicals inhibiting BC progression by regulating the DNA methylation pattern on the ncRNA promoter. For example, Jiang et al. suggested that glabridin up-regulates miR-148a expression through DNA promoter demethylation and inhibits the expression of SMAD2, restoring epithelial-like characteristics of BC cells and reducing stem-like properties [124]. Similarly, Kim et al. demonstrated that ginsenoside Rg3 downregulates oncogenic lncRNA ATXN8OS expression by mediating hypermethylation of the promoter, thus inhibiting the proliferation of BC cells [125]. Moreover, the phytochemicals can also regulate the histone methylation pattern of ncRNA to inhibit breast tumor progression. Han et al. revealed that treating BC cells with delphinidin can reduce methylation marker H3K27me3 on the miR-34a promoter, minimizing the inhibitory effect of H3K27me3 on miR-34a, activating miR-34a expression and inhibiting BC occurrence [126]. Relevant studies of phytochemical-mediated ncRNA methylation pattern alterations are summarized in Table 4.

Arsenic trioxide, metformin, and decitabine inhibit BC progression by modulating ncRNA promoter methylation [130,131,132] (Table 4). Decitabine is a methyltransferase DNMT1 inhibitor that can be applied to various cancer therapies [133]. Fu et al. indicated that decitabine-treated BC cells activate the expression of lncRNA PAS1 and inhibit growth and metastasis through the DNMT1/PAS1/PH20 axis [132]. Those findings suggest that decitabine can restore lncRNA expression by inhibiting methyltransferase. It is speculated that methyltransferase inhibitors may restore the activity and expression of other ncRNAs, which can be inhibited by hypermethylation.

The major potential danger of using methyltransferase inhibitors to treat cancer is that the drugs are not specific to treatment, which may cause immeasurable harm to the human body and lead to severe side effects. However, the cytotoxicity of treatment by specific miRNA oligonucleotides targeting methyltransferases might be less than methyltransferase inhibitors. MiRNA oligonucleotides may induce epigenetically silenced tumor suppressor genes to recover expression by targeting methyltransferases. In addition, on consideration of the advantage of natural phytochemicals acting as epigenetic therapeutic drugs, it is apparent that these substances specifically attenuate the malignant behavior of cancer cells but usually have no harmful effects on the normal cells. Therefore, it is feasible and preferable to combine phytochemicals and ncRNA oligonucleotide drugs targeting epigenetic regulators to treat tumors in a gentle and specific manner in the future. Additionally, each key target of epigenetic drugs to treat BC should be adequately considered to avoid disorders to achieve optimal therapeutic effects.

### 3.2. Diagnostic Markers Based on Methylation of ncRNA

Abnormal epigenetic changes of ncRNA and aberrant ncRNA dysregulation led to disparate malignant biological behavior of BC (Figure 4). Therefore, these aberrant epigenetic targets have potential as diagnostic markers. MiRNA and lncRNA gene DNA methylation status in PB leukocytes can serve as a potential diagnostic marker in BC. For instance, a prospective study on miRNA gene methylation suggests that there are five differentially methylated miRNA genes in PB of patients before a diagnosis of BC, indicating that the abnormal methylation of miRNA genes may be an event before BC initiation and differential methylation of miRNA genes can be used as risk biomarkers for BC [134]. Furthermore, another prospective study using digital PCR to detect DNA in PB leukocytes shows that hypermethylation of the LINC00299 gene is an effective biomarker of TNBC in young women [135]. These findings shed light on the fact that the methylation pattern of ncRNAs can be used as a molecular diagnostic marker of BC.

The concept of single or multiple ncRNA methylation statuses as diagnostic markers of BC is prevalent in patients. Until now, there have been numerous proposed candidate ncRNA methylation markers, with no biomarkers being widely used in the current clinical diagnosis practice of BC [136,137]. The advantages of the diagnostic marker system based on ncRNA methylation status are: (1) epigenetic biomarkers may be more sensitive than genes in predicting precancerous lesions; (2) the advantages of using PB as a humoral sample include simple detection, repeatability, and non-invasion biomarkers; (3) epigenetic biomarkers are more stable in body fluids than RNA or cell-free DNA biomarkers; (4) epigenetic biomarkers are more specific to tumor organs and tissues. Applying ncRNA in clinical diagnosis still requires a lot of effort and other questions must be solved urgently in the future. The development of more accurate and sensitive assays is needed to detect ncRNA methylation status and epigenetic biomarkers must be validated in large and reliable cohorts.

## 4. Summary

Comprehensively summarized findings about the function of mutual regulation of ncRNAs and methylation in BC progression in this review further strengthen our deep understanding of epigenetic mechanisms in BC development. The role of complicated links between ncRNAs and methylation in promoting the malignant progression of BC is discussed in great detail and the potential of ncRNA methylation modifications as diagnostic markers and therapeutic targets is highlighted. Methylation modifications mainly regulate ncRNA expression by altering the DNA promoter CpG island methylation state, the m6A modification of naive ncRNAs, and histone methylation modification patterns, while ncRNAs mainly transform methylation patterns via regulation of methylation modification-related enzymes or recruiting chromatin-modifying factors [29,61,76,83,100] (Figure 2). The disorder of the interaction between ncRNAs and methylation modifications can promote proliferation, metastasis, stem cell-like properties, drug resistance, autophagy, and other malignant behaviors conducive to breast tumor progression [29,44,50,51,63,96,109] (Figure 4, Table 1, Table 2 and Table 3).

Epigenetic regulation can predict the possibility of BC occurrence in advance, reflecting the advantages of epigenetics in precancerous warning and early diagnosis. In addition, the epigenetic mechanism is reversible while genetic regulation is irreversible, therefore, one can develop drugs targeting reversible processes to treat BC [138]. Considering taking epigenetic regulation-related enzymes as drug targets to reshape epigenetic dynamics and restore general gene expression would be an ideal therapy strategy for BC [133]. Notably, epigenetic modifications are complex and diverse, with synergistic effects. The tight regulation of various epigenetic mechanisms on the occurrence and development of BC needs to be explored in depth. Further exploration of the correlation between ncRNA and methylation will help us determine the complicated epigenetic regulatory network and discover new therapeutic targets for BC oncogenesis.

## Figures and Tables

**Figure 1 ijms-23-15759-f001:**
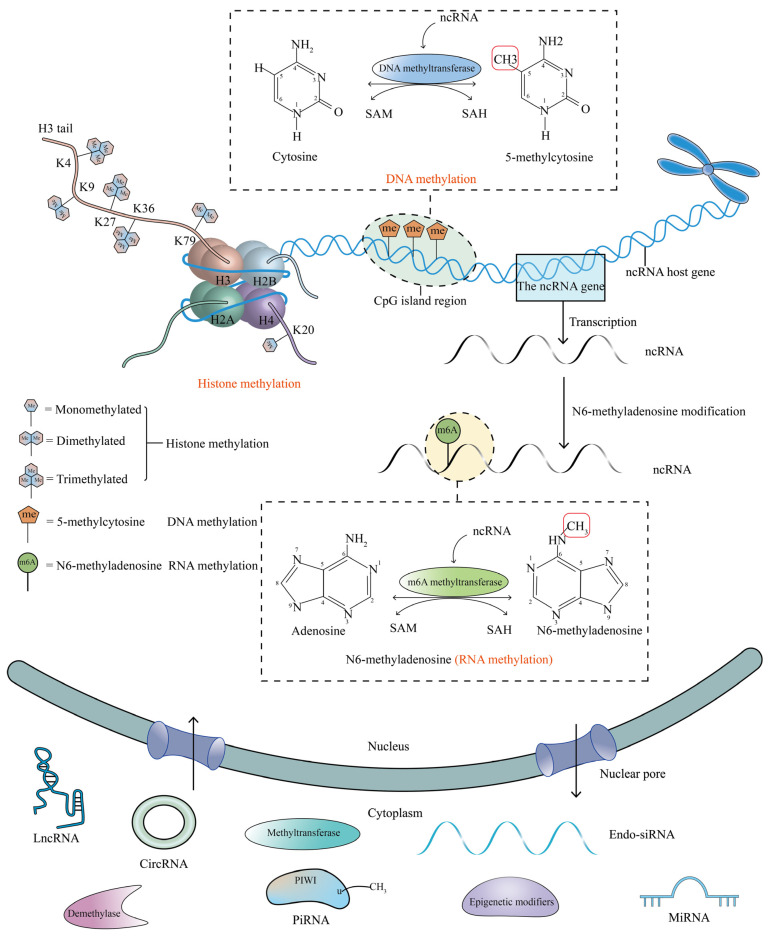
DNA methylation of the ncRNA host gene, m6A modification of ncRNA, and chromatin regulation by histone modifications. DNA methylation refers to a kind of methylation modification on DNA, which is under the action of DNA methyltransferase, S-adenosyl methionine (SAM) acts as a methyl donor to covalently add a methyl group at the cytosine 5 carbon position of the CpG dinucleotide in the genome, and SAM is transformed into SAH (S-adenosylhomocysteine). N6-methyladenosine is a type of RNA methylation modification which is the m6A methyltransferase that adds a methyl group to the N6 position of adenosine, converting SAM to SAH. Histones are the core components of the nucleosome subunits in which histones H3, H4, H2A, and H2B form an octamer surrounded by DNA fragments. Histone methylation is a covalent modification that usually occurs on H3 and H4 lysine residues, which can be monomethylated, dimethylated, even trimethylated. K4, K9, K27, K36, and K79 of H3 and K20 of histone H4 are common sites of histone lysine methylation. These catalytic processes are reversible, and the red box represents the added methyl group. The ncRNAs undergoing chromatin modification, DNA methylation modification, and m6A modification, are transferred from the nucleus to the cytoplasm through the nuclear pores, and then form mature ncRNAs in the cytoplasm. The mature ncRNAs can regulate the protein expression levels of methylation modifiers in the cytoplasm and can also enter the nucleus through the nuclear pore to regulate the function of methylation modifiers.

**Figure 2 ijms-23-15759-f002:**
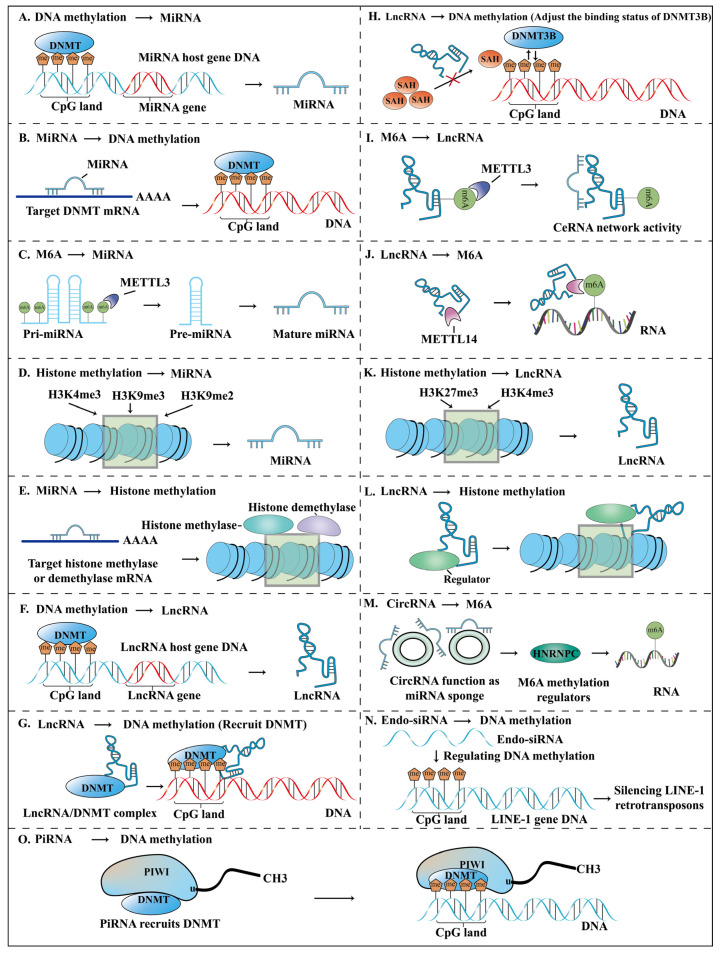
The summary of the interaction between ncRNAs and methylation modifications in BC. (**A**) Regulation of miRNA by DNA methylation. (**B**) Regulation of DNA methylation by miRNA. (**C**) Crosstalk between m6A and miRNA. (**D**) Regulation of miRNA by histone methylation. (**E**) Regulation of histone methylation by miRNA. (**F**) Regulation of lncRNA by DNA methylation. (**G**). Regulation of DNA methylation by lncRNA (lncRNA recruits DNMTs). (**H**) Regulation of DNA methylation by lncRNA (lncRNA regulates the binding status of DNMTs). (**I**) m6A modification on lncRNA. (**J**) lncRNA regulates m6A RNA methylation. (**K**) Regulation of lncRNA by histone methylation. (**L**) Regulation of histone methylation by lncRNA. (**M**) CircRNA indirectly regulates m6A modification. (**N**) Regulation of DNA methylation by endo-siRNA. (**O**) Regulation of DNA methylation by piRNA.

**Figure 3 ijms-23-15759-f003:**
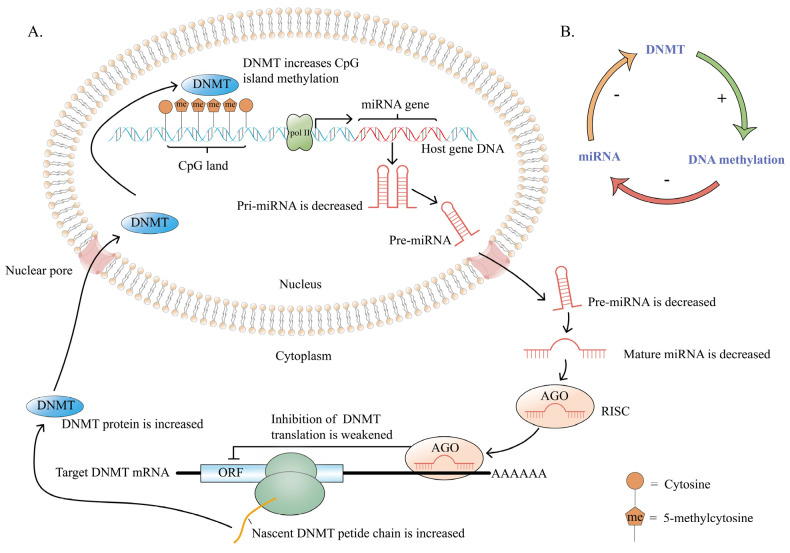
The miRNA/DNMT regulatory loop. (**A**) The methylation state of the miRNA host gene promoter is regulated by DNA methyltransferase; miRNAs can target and negatively regulate DNA methyltransferase expression in turn, which can form a regulatory loop involved in the initiation and development of BC. DNMT acts on the CpG island region of the miRNA host gene promoter, causing hypermethylation and inhibiting pri-miRNA synthesis. Pri-miRNA generates pre-miRNA after continuous shearing and exports it to the cytoplasm. Next, pre-miRNA is cleaved to generate mature miRNA. The miRNA and Argonaute (AGO) form a transcriptional silencing complex, which weakens the inhibitory effect on the target DNMT mRNA, subsequently increasing the DNMT protein. Then, DNMT is transported into the nucleus where it increases the methylation degree of CpG islands and further inhibits miRNA expression. A miRNA/DNMT regulatory circuit is formed. (**B**) A brief schematic diagram of the miRNA/DNMT regulatory circuit. Pri-miRNA, primary-miRNA. Pre-miRNA, precursor-miRNA. RISC, RNA-induced silencing complex. ORF, open reading frame. DNMT, DNA methyltransferase.

**Figure 4 ijms-23-15759-f004:**
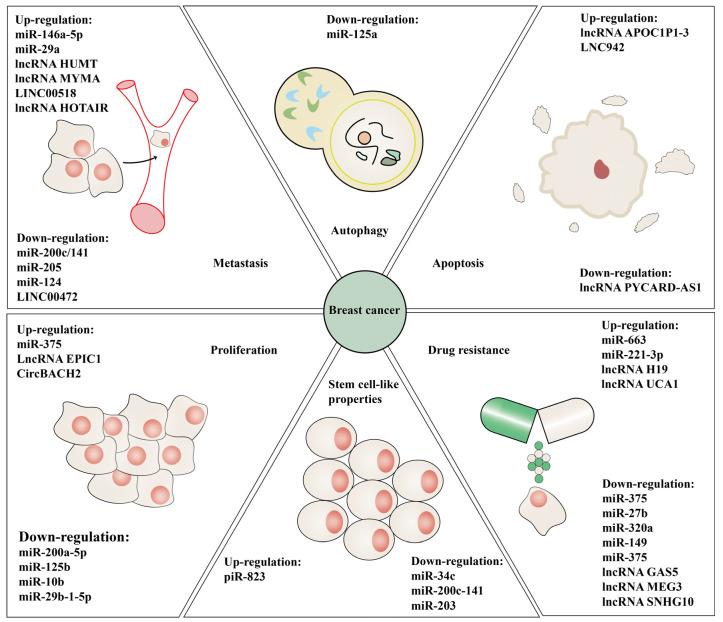
MiRNA, lncRNA, circRNA, and piRNA are associated with proliferation, apoptosis, metastasis, autophagy, stemness, and drug resistance in BC. Dysregulation of ncRNAs due to aberrant crosstalk between methylation modification and ncRNAs is associated with malignant behavior in BC, such as proliferation (miR-375, lncRNA EPIC1, circBACH2. miR-200a-5p, miR-125b, miR-10b, miR-29b-1-5p), apoptosis (lncRNA APOC1P1-3, LNC942. lncRNA PYCARD-AS1), metastasis (miR-146a-5p, miR-29a, lncRNA HUMT, lncRNA MYMA, LINC00518, and lncRNA HOTAIR. miR-200c/141, miR-205, miR-124, and LINC00472), autophagy (miR-125a), stemness (piR-823, miR-34c, miR-200c-141, and miR-203), and drug resistance (miR-663, miR-221-3p, lncRNA H19, lncRNA UCA1. miR-375, miR-27b, miR-320a, miR-149, miR-375, lncRNA GAS5, lncRNA MEG3, lncRNA SNHG10).

**Table 1 ijms-23-15759-t001:** Crosstalk between different methylation types and miRNAs in BC.

MiRNAs	Epigenetic Mechanism	Expression in BC	Target Gene or Protein/Pathway Involved	Biological Function	Ref.
MiR-362-3p miRNA-329	Regulation of miRNA expression by DNA methylation	↓	MiR-362-3p/miRNA-329- *p130Cas*	Inhibiting proliferation, migration, and invasion	[29]
MiR-195/497		↓	MiR-195-497/Raf-1	Suppressing cell proliferation and invasion	[30]
MiR-892b		↓	MiR-892b/NF-κB	Inhibiting aggressiveness	[31]
MiR-200b		↓	Kindlin 2/DNMT3A/miRNA-200b	Inhibiting invasion	[32]
MiR-200a-5p		↓	FEN1/PCNA/DNMT3a/miR-200a-5p	Inhibiting proliferation	[33]
MiR-193a-3p		↓	MiR-193a-3p/*GRB7*	Inhibiting HER2 positive BC aggressiveness	[34]
MiR-205		↓	Mel-18/DNMT/miR-205	Inhibiting EMT	[35]
MiR-34 miR-199a/b		↓	MiR-34-miR-199a/b/Axl	Inhibiting migration and invasion	[36]
MiR-335		↓	Not applicable	Inhibiting tumor reinitiation	[37]
MiR-126 miR-126 ^∗^		↓	MiR-126-miR-126 ^∗^/SDF-1α	Inhibiting metastasis	[38]
MiR-100		↓	MiR-100/*SMARCA5* MiR-100/*HOXA1*	EMT-inducing and tumor-suppressing effects	[39]
MiR-125b		↓	MiR-125b/*ETS1*	Suppressing proliferation	[40]
MiR-10b^∗^		↓	MiR-10b^∗^/*BUB1*, *PLK1* and *CCNA2*	Inhibiting proliferation	[41]
MiR-34c		↓	MiR-34c/*Notch4*	Inhibiting tumor-initiating cells self-renewal	[42]
MiR-200c-141		↓	Not applicable	Decreasing stem-like properties	[43]
MiR-663		↑	MiR-663/*HSPG2*	Inducing chemotherapy resistance	[44]
MiR-320a		↓	MiR-320a/*TRPC5* and *NFATC3*	Decreasing chemoresistance	[45]
MiR-149		↓	MiR-149/*NDST1*	Modulating chemoresistance	[46]
MiR-375		↓	MiR-375/*IGF1R*	Inhibiting trastuzumab resistance	[47]
MiR-29c		↓	Not applicable	Involving in invasion and chemotherapy sensitivity	[48]
MiR-9		↓	DNMT3A/miR-9/*VEGFA*	Regulating mechanical compression	[49]
MiR-200c		↓	MiR-21a/miR-200c/*PTEN*	Inhibiting the transformation of M2 macrophages	[50]
MiR-125a		↓	Not applicable	Reducing autophagy	[51]
MiR-148b MiR-29c MiR-26b	Regulation of DNA methylation by miRNA	↓	MiRNAs/DNMT3B	Not applicable	[52]
MiR-29c		↓	MiR-29c/DNMT3B/TIMP3/STAT1/FOXO1	Inhibiting aggressiveness	[53]
MiR-29b-1-5p		↓	MiR-29b-1-5p/DNMTs	Inhibiting cell proliferation	[54]
MiR-770-5p		↓	MiR-770-5p/DNMT3A	Inhibiting EMT	[55]
MiR-148a/152		↓	MiR-148a/152/DNMT1 regulatory circuit	Inhibiting aggressiveness	[56]
MiR-200b		↓	MiR-200b/DNMT3A regulatory circuit	Inhibiting aggressiveness	[57]
MiR-133a-3p		↓	MiR-133a-3p/MAML1/DNMT3A regulatory circuit	Inhibiting migration and stemness	[58]
MiR-10a		↑	MiR-10a/*Hoxd4*	Not applicable	[59]
MiR-146a-5p	M6A modification of miRNA	↑	METTL14/miR-146a-5p	Promoting migration and invasion	[60]
MiR-221-3p		↑	METTL3/miR-221-3p/HIPK2/Che-1	Inducing chemotherapy resistance	[61]
MiR-375	Regulation of miRNA expression by histone methylation	↑	MiR-375/RASD1	Promoting cell proliferation	[62]
MiR-200c		↓	MiR-200c/ZEB1	Inhibiting lymph node metastasis	[63]
MiR let-7e		↓	JARID1B/let-7e/cyclin D1	Inhibiting proliferation	[64]
MiR-7	Regulation of histone methylation by miRNA	↓	MiR-7/SET8	Inhibiting EMT and invasiveness	[65]
MiR-491-5p		↓	MiR-491-5p/JMJD2B	Inhibiting proliferation	[66]
MiR-29a		↑	MiR-29a/SUV420H2	Promoting EMT, migration, and invasion	[67]

EMT, epithelial–mesenchymal transition. ↓, down-regulation in BC. ↑, up-regulation in BC. *: In the earlier miRNA nomenclature, miRNAs with higher expression levels are not followed by any symbols, while miRNAs with lower expression levels are followed by “*”, such as miR-126* and miR-10b* in the table.

**Table 2 ijms-23-15759-t002:** Crosstalk between different methylation types and lncRNAs in BC.

LncRNAs	Epigenetic Mechanism	Expression in BC	Target Gene or Protein/Pathway Involved	Biological Function	Ref.
LncRNA APOC1P1-3	Regulation of lncRNA expression by DNA methylation	↑	APOC1P1-3/α-tubulin acetylation	Inhibiting apoptosis	[75]
LncRNA EPIC1		↑	LncRNA EPIC1/MYC	Promoting Cell-Cycle progression	[76]
LncRNA HUMT		↑	HUMT/YBX1/FOXK1 axis	Promoting lymphangiogenesis and metastasis	[77]
LncRNA GAS5		↓	Not applicable	Promoting chemosensitivity and apoptosis	[78]
LncRNA MEG3		↓	Not applicable	Promoting chemosensitivity	[79]
LncRNA MYMA	Regulation of DNA methylation by lncRNA	↑	ROR1/HER3/lncRNA MYMA axis	Promoting bone metastasis	[80]
LncRNA 91H		↑	LncRNA 91H *-H19/IGF2*	Increasing aggressive phenotype	[81]
LncRNA PYCARD-AS1		Not applicable	LncRNA PYCARD-AS1/*PYCARD*	Regulating apoptosis	[82]
LINC00518		↑	LINC00518/*CDX2*/Wnt signaling	Promoting epithelial cell growth and metastasis	[83]
LncRNA H19		↑	H19/SAHH/DNMT3B axis	Promoting tamoxifen resistance	[84]
LncRNA BCLIN25		↑	BCLIN25/miR-125b/ERBB2 axis	Promoting tumorigenesis	[85]
LncRNA MIAT		↑	LncRNA MIAT/*DLG3/*Hippo signaling pathway	Promoting tumorigenesis	[86]
LINC00472		↓	LINC00472/*MCM6*/MEK/ERK signaling pathway	Inhibiting metastasis	[87]
LncRNA TINCR		↑	STAT3/TINCR/EGFR feedback loop	Promoting tumorigenesis	[88]
LncRNA LINC00922		↑	LINC00922/*NKD2*/Wnt signaling pathway	Promoting EMT, invasive and migratory capacities	[89]
LncRNA MAGI2-AS3		↓	MAGI2-AS3/*MAGI2*/Wnt/β-catenin pathway	Inhibiting cell proliferation and migration	[90]
LncRNA SNHG10		↓	LncRNA SNHG10/miR-302b	Promoting chemosensitivity of TNBC cells to doxorubicin.	[91]
LNC942	Regulation of m6A modification by lncRNA	↑	LNC942/METTL14/CXCR4 and CYP1B1 signaling axis	Promoting proliferation and inhibiting apoptosis	[92]
LINC00675	M6A modification of lncRNA	↓	METTL3/LINC00675/miR-513b-5p	Suppressing proliferation, migration, and invasion	[93]
LncRNA DLEU1	Regulation of lncRNA expression by histone methylation	↑	Not applicable	Promoting tumorigenesis	[94]
LncRNA EPB41L4A-AS2		↓	ZNF217/EPB41L4A-AS2/RARRES1	Inhibiting proliferation, migration, and invasion and inducing apoptosis	[95]
LncRNA HOTAIR		↑	Not applicable	Associating with endocrine disruption	[96]
LncRNA UCA1		Not applicable	SATB1/lncRNA UCA1	Promoting growth and metastasis	[97]
LncRNA HOTAIR	Regulation of histone methylation by lncRNA	↑	Not applicable	Promoting invasiveness and metastasis	[98]
LncRNA UCA1		↑	LncRNA UCA1/EZH2/p21 axis	Inducing tamoxifen resistance	[99]
LINC00511		↑	LINC00511/EZH2/*CDKN1B*	Promoting proliferation and inhibiting apoptosis	[100]
LncRNA ROR		↑	LncRNA ROR/MLL1/TIMP3	Promoting BC progression	[101]
LncRNA PHACTR2-AS1		↓	EZH2/lncRNA PHACTR2-AS1	Inhibiting growth and metastasis	[102]

EMT, epithelial–mesenchymal transition. ↓, down-regulation in BC. ↑, up-regulation in BC.

**Table 3 ijms-23-15759-t003:** Crosstalk between different methylation types and other ncRNAs in BC.

NcRNAs	Epigenetic Mechanism	Expression in BC	Target Gene or Protein/Pathway Involved	Biological Function	Ref.
CircBACH2	CircRNA indirectly regulates m6A modification	↑	CircBACH2/hsa-miR-944/HNRNPC axis	Promoting cell proliferation	[106]
Endo-siRNA	Regulation of DNA methylation by endo-siRNA	↓	Endo-siRNA/LINE-1	Maintaining a stable genome structure	[107]
PiRNA-651	Regulation of DNA methylation by piRNA	↑	PIWI/piRNA-651/DNMT1/*PTEN*	Promoting cell proliferation and migration and inhibiting apoptosis	[108]
PiR-823		↑	PiR-823/DNMT/*APC*/Wnt signaling	Promoting cell proliferation and being involved in stem cell regulation	[109]

↓, down-regulation in BC. ↑, up-regulation in BC.

**Table 4 ijms-23-15759-t004:** Treatment strategies based on methylation of ncRNAs.

ncRNA	Therapeutic Substance	Epigenetic Mechanism	Target Gene or Protein/Pathway Involved	Biological Function	Ref.
MiR-148a	Glabridin	DNA methylation	MiR-148a/TGFβ/SMAd2	Stemness	[124]
LncRNA ATXN8OS	Ginsenoside Rg3	DNA methylation	LncRNA ATXN8OS/miR-424-5p	Proliferation	[125]
MiR-34a	Delphinidin	H3K27me3	HOTAIR/miRNA-34a	Carcinogenesis	[126]
MiR-34a	3,6-Dihydroxyflavone	DNA methylation	PI3K/Akt/mTOR	Carcinogenesis	[127]
MiR-216a-3p	Limonin	DNA methylation	MiR-216a-3p/*WNT3A*	Stemness	[128]
LncRNA C3orf67-AS1	Ginsenoside Rh2	DNA methylation	Not applicable	Proliferation	[129]
LncRNA HOTAIR	Metformin	DNA methylation	Not applicable	EMT	[130]
miR-200c	Arsenic trioxide	DNA methylation	Not applicable	Migration and invasion	[131]
LncRNA PAS1	Decitabine	DNA methylation	DNMT1/PAS1/PH20	Growth and metastasis	[132]

EMT, Epithelial–mesenchymal transition. H3K27me3, trimethylated histone H3 at lysine 27.

## Data Availability

Not applicable.

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
