# Peer review of "Crosstalk between Methylation and ncRNAs in Breast Cancer: Therapeutic and Diagnostic Implications"

_ijms, 2022, doi:10.3390/ijms232415759_

Round 1

Reviewer 1 Report

The review makes a comprehensive summary of the interaction between ncRNA and significant methylation modification in Breast Cancer. The ncRNA discussed included miRNA, lncRNA, cirRNA, endo-siRNA and piRNA. The methylation modification includes DNA methylation, m6A, and histone methylation. Because of the prevalence of breast cancer and the growing attention to epigenetic regulation, the paper has focused on an important area. The author provided figures that help to clearly explain the complicated mechanism. The review is overall well organized, with some detail to be improved.

  1. In page 10, “Regulation of DNA methylation by lncRNA ” part, the author indicates that lncRNA can recruit DNMT and adjust the binding status of DNA methyltransferase. What are the differences between these two? An additional figure will help to clarify.
  2. In figure 2, the author kindly provided figures for all the situation that is discussed. However, it’s hard to match the figure with the situation.

a.       A subtitle for each panel is recommended. A simple title like “lncRNA--> DNA methylation” will make it easier to read.

b.      In the later paragraph, figure 2 can be referred to to make the mechanism more straightforward. 

3.      The format of the reference needs to be updated. There are warning for “reference source not found”.

  1. Additional lines are needed for the tables for each RNA. It’s hard to read with so many RNAs

5.      For Figure 4, it seems these ncRNAs are working as biomarkers. The title needs to indicate that. For example, “Circulating ncRNA associated with malignant behavior in BC”.

6.      Table2, lncRNASNHG10 part, is doxorubicin any different from other chemotherapy? If not the biological function should be promoting chemosensitivity

Author Response

Thank you for your comments concerning our manuscript entitled “Crosstalk between methylation and ncRNAs in breast cancer: therapeutic and diagnostic implications” (Manuscript ID: ijms-2047585). Those comments are all valuable and greatly helpful for revising and improving our paper. We have studied the comments carefully and have made a correction, which we hope meets with approval. Revised portions are marked in red on the paper. The main corrections in the paper and the responses to the reviewer’s comments are as follows:

Point 1: In page 10, “Regulation of DNA methylation by lncRNA ” part, the author indicates that lncRNA can recruit DNMT and adjust the binding status of DNA methyltransferase. What are the differences between these two? An additional figure will help to clarify.

Response 1: We greatly appreciate your insightful comment. Indeed, as the reviewer suggested, there are differences between “recruit DNMT” and “adjust the binding status of DNMT.” The mechanism by which “lncRNA recruits DNMT” is that lncRNA interacts with DNA methyltransferases to regulate DNA methyltransferase localization in the gene promoter region. However, the mechanism of “adjustment of the binding status of DNMT” is that it adjusts the binding state of DNMT and the promoter region by controlling the accumulation of SAH (which is produced by the demethylation of the methylating substrate SAM). We carefully revised the corresponding text and Figure 2 according to the reviewer’s suggestions. A panel was added in Figure 2H to clarify and explain the exact mechanism of lncRNA regulating the binding state of DNA methyltransferase so that we can distinguish the mechanism of lncRNA recruiting DNA methyltransferase. We also made minor modifications in the main texts to distinguish the difference between the two regulation mechanisms (page 10, lines 288–291), uploaded the changed diagram (page 6, line 157), and made minor adjustments to the figure legends and reference order as required.

Point 2: In figure 2, the author kindly provided figures for all the situation that is discussed. However, it’s hard to match the figure with the situation.

  1. A subtitle for each panel is recommended. A simple title like “lncRNA--> DNA methylation” will make it easier to read.
  2. In the later paragraph, figure 2 can be referred to to make the mechanism more straightforward.

Response 2: We appreciate the professional comments the reviewer provided. We have made corrections according to the reviewer’s suggestion. We have added subtitles for each panel, like “lncRNA--> DNA methylation” and renewed the diagram in the texts (page 6, line 157). We also refer to Figure 2 in the appropriate paragraph position to make the mechanism more direct and straightforward so that readers can easily match the figure with contexts.

Point 3: The format of the reference needs to be updated. There are warning for “reference source not found”.

Response 3: Thank you for the helpful comments. We carefully compared and checked the reference formats of journals and found that there were indeed some minor differences. The style of the references in the whole manuscript has been corrected according to journal requirements.

Point 4: Additional lines are needed for the tables for each RNA. It’s hard to read with so many RNAs.

Response 4: We thank the reviewer for pointing out this issue. We have added extra lines to make it easier for readers. The format of the tables has been modified to ensure that each ncRNA is readable.

Point 5: For Figure 4, it seems these ncRNAs are working as biomarkers. The title needs to indicate that. For example, “Circulating ncRNA associated with malignant behavior in BC”.

Response 5: Thank you for the suggestion. We apologize that we previously neglected to indicate the role of these ncRNAs as biomarkers in the title of Figure 4. Now so we have changed the title to “MiRNA, lncRNA, circRNA, and piRNA are associated with proliferation, apoptosis, metastasis, autophagy, stemness, and drug resistance in BC” and marked the malignant biological behavior corresponding to each ncRNA in the figure legend (page 14, lines 467–480).

Point 6: Table2, lncRNASNHG10 part, is doxorubicin any different from other chemotherapy? If not the biological function should be promoting chemosensitivity.

Response 6: We greatly appreciate the comment and have corrected it. To the best of our knowledge, actually, there is no difference between doxorubicin and other chemotherapy drugs. Therefore, we have revised the biological function to “promoting chemosensitivity” in Table 2.

Reviewer 2 Report

It is acceptable in current form.

Author Response

We are extremely grateful to you for affirming the content of our manuscript entitled “Crosstalk between methylation and ncRNAs in breast cancer: therapeutic and diagnostic implications” (Manuscript ID: ijms-2047585). A special thanks to you for your positive comments.

Reviewer 3 Report

The review describes well about the crosstalk between methylation and ncRNAs in breast cancer. It looks interesting.  

Author Response

We greatly appreciate your kind comment concerning our manuscript entitled “Crosstalk between methylation and ncRNAs in breast cancer: therapeutic and diagnostic implications” (Manuscript ID: ijms-2047585). We would like to express our great appreciation to you for your positive comments.
